# The antibiotic resistance reservoir of the lung microbiome expands with age in a population of critically ill patients

Victoria T. Chu [ID][1,2], Alexandra Tsitsiklis[2], Eran Mick [ID][2,3,4], Lilliam Ambroggio[5], Katrina L. Kalantar[6], Abigail Glascock[3], Christina M. Osborne[5], Brandie D. Wagner[5,7], Michael A. Matthay [ID][4], Joseph L. DeRisi[3,8], Carolyn S. Calfee[4], Peter M. Mourani [ID][9] & Charles R. Langelier [ID][2,3] ✉

Antimicrobial resistant lower respiratory tract infections are an increasing public health threat and an important cause of global mortality. The lung microbiome can influence susceptibility of respiratory tract infections and represents an important reservoir for exchange of antimicrobial resistance genes. Studies of the gut microbiome have found an association between age and increasing antimicrobial resistance gene burden, however, corollary studies in the lung microbiome remain absent. We performed an observational study of children and adults with acute respiratory failure admitted to the intensive care unit. From tracheal aspirate RNA sequencing data, we evaluated age-related differences in detectable antimicrobial resistance gene expression in the lung microbiome. Using a multivariable logistic regression model, we find that detection of antimicrobial resistance gene expression was significantly higher in adults compared with children after adjusting for demographic and clinical characteristics. This association remained significant after additionally adjusting for lung bacterial microbiome characteristics, and when modeling age as a continuous variable. The proportion of adults expressing beta-lactam, aminoglycoside, and tetracycline antimicrobial resistance genes was higher compared to children. Together, these findings shape our understanding of the lung resistome in critically ill patients across the lifespan, which may have implications for clinical management and global public health.

Antimicrobial resistance (AMR) is one of the top global health threats facing humanity[1]. Lower respiratory tract infections (LRTI) are a leading cause of death worldwide[1,2], and account for a disproportionate burden of global AMR-related mortality with an estimated 1.5 million deaths in 2019 attributable to resistant microbes[2].

Despite the rise in resistant respiratory infections, the antimicrobial resistance genes (ARG) within the lung microbiome remain understudied and incompletely defined[3]. As with the gastrointestinal tract, the respiratory tract harbors diverse microbial communities acquired early during life[4–6] that are continually influenced over the lifespan by

[1]Division of Infectious Diseases & Global Health, University of California, San Francisco, CA, USA. [2]Division of Infectious Diseases, University of California, San Francisco, CA, USA. [3]Chan Zuckerberg Biohub, San Francisco, CA, USA. [4]Division of Pulmonary and Critical Care Medicine, Cardiovascular Research Institute, University of California, San Francisco, CA, USA. [5]Department of Pediatrics, University of Colorado and Children's Hospital Colorado, Aurora, CO, USA. [6]Chan Zuckerberg Initiative, San Francisco, CA, USA. [7]Department of Biostatistics and Informatics, Colorado School of Public Health, University of Colorado, Aurora, CO, USA. [8]Department of Biochemistry and Biophysics, University of California, San Francisco, CA, USA. [9]Arkansas Children's Research Institute, Arkansas Children's Hospital, Little Rock, AR, USA. ✉e-mail: chaz.langelier@ucsf.edu

exposures to organisms from the environment and other humans, as well as to antimicrobials. The gut, respiratory tract, and other human anatomical microbiomes serve as reservoirs for ARGs, or antimicrobial resistomes, and act as potential sites of ARG acquisition and transmission[7].

An understanding of the epidemiological, biological, and clinical factors associated with AMR acquisition is crucial to halting the spread of resistant infections. Prior studies of the gut microbiome have demonstrated an association between age and the composition and burden of ARGs[8,9], suggesting that cumulative exposures might shape the resistance landscape of endogenous microbial communities. Other factors influencing the gut resistome include travel[10], hospital exposure[11], and antibiotic use[12]. Despite these findings, corollary studies in the respiratory microbiome have not yet been performed, a key gap given the global magnitude of drug resistant LRTI. Furthermore, few studies have used metatranscriptomic RNA sequencing (RNA-seq) to both profile lower respiratory microbial ecology and detect ARG expression in the airway microbiome[3,13]. Prior work suggests that RNA-seq may have higher sensitivity and specificity for ARG detection compared with DNA metagenomic sequencing[3]. Additionally, while DNA metagenomic sequencing captures the potential ARG reservoir of the lung microbiome, RNA sequencing allows for assessment of the actively transcribed component of this reservoir.

Here, we sought to test the hypothesis that older age is associated with an increased prevalence of ARGs in the lung microbiome, using metatranscriptomics and multivariable logistic regression modeling. We find that age is indeed an independent risk factor for detecting ARGs in the lower airway microbiome, even after adjusting for multiple covariates including sex, race/ethnicity, LRTI diagnosis, community-versus hospital-acquired infection, days from intubation to specimen collection, and composition of the lung microbiome.

## Results

### Patient Cohorts

We studied 261 children (median age: 1 year, interquartile range (IQR): 0–15 years, range: 0–17 years), and 88 adults (median age: 63 years, IQR: 54–72 years, range: 21–94 years) (Supplemental Table 1). Of the 349 patients, 231 (66%) were adjudicated as LRTI-positive, 67 (19%) had no evidence of LRTI, and 51 (15%) of patients had indeterminate LRTI status. The proportion of patients in each LRTI adjudication group did not differ between the two cohorts. Adults had a higher proportion of hospital-acquired LRTI (HA-LRTI) than children (25% vs. 6%, respectively), emphasizing the need to include this as a covariate in our subsequent logistic regression model. In both cohorts, 90% of the patients received antibiotics prior to tracheal aspirate collection. All four U.S. census regions (Midwest, Northeast, South, West) in the U.S. were represented among the 261 pediatric patients; adult patients were from one enrollment site located in the regional West.

Median total reads per sample for adults was 57.55 million reads (IQR: 47.16–78.19 million reads); medial total reads per sample for children was 80.53 million reads (IQR: 54.09–123.95 million reads). Reads and average read depth, normalized for ARG length, per ARG were a median of 4.00 reads (IQR: 2.00–8.25 reads) and 1.00x read depth (IQR: 0.99–1.98x read depth), respectively, for adults, and a median of 4.00 reads (IQR: 2.00–14.50 reads) and 1.63x read depth (IQR: 0.99–2.94x read depth), respectively, for children.

### Lower Respiratory Tract Resistome

ARGs were detectably expressed in the lower respiratory tract microbiome of 40 (45%) adults compared with 53 (20%) children (Pearson's Chi-square $p < 0.01$). Across all patients, 74 distinct ARGs representing nine ARG classes were detected (Fig. 1A). The number of detectably expressed ARGs (Fig. 1B) and the number of ARG classes (Fig. 1C) significantly differed between the youngest age subgroups (0–2 years and 3–10 years) and the oldest age subgroups (60–69 years, 70–79 years, and ≥80 years age groups), respectively. A significant increase was also noted between the 3–10 and the 11–18 years of age subgroups.

The most frequently detected ARG classes across all patients conferred resistance to beta-lactams ($n = 85$ patients), macrolides-lincosamide-streptogramin (MLS) ($n = 41$), and aminoglycosides ($n = 37$). A greater proportion of adults compared with children had expression beta-lactam ARGs (31% (95% confidence interval (CI): 21–41%) vs 13% (95% CI: 10–18%)), aminoglycoside ARGs (20% (95% CI: 13–30%) vs. 2% (95% CI: 0.6–4%)), and tetracycline ARGs (14% (95% CI: 7–23%) vs 3% (95% CI: 1–5%)) (Fig. 1D, Supplemental Table 2). When evaluated by age subgroup, the proportion of patients with detectable expression of beta-lactam, MLS, or tetracycline ARGs was highest in patients ≥70 years of age (Supplemental Fig. 1). Among young children, 13% (95% CI: 8–19%) of patients aged 0–2 years had expressed a beta-lactam ARG compared with 2% (95% CI: 0.05–10%) of patients aged 3–10 years; this pattern was not seen for the other ARG classes. Among the beta-lactam ARGs, we detected six *AmpC* beta-lactamase genes, five extended-spectrum beta-lactamase genes, and two carbapenemase genes (Supplemental Fig. 3).

ARG alpha diversity was higher in adults than children as measured by the Shannon Diversity Index, and increased primarily in patients ≥60 years of age (Supplemental Fig. 2). The composition of the lung resistome significantly differed between children and adults, as measured by the Bray Curtis dissimilarity index ($p = 0.005$ by PERMANOVA) (Fig. 1E).

In a logistic regression model assessing the association of binary age group with detection of any ARG expression, accounting for sex, race/ethnicity, LRTI status (community-acquired CA-LRTI), HA-LRTI, No LRTI), and days from intubation to specimen collection, the risk of ARG detection was increased in adults compared with children (adjusted odds ratio [aOR]: 2.16, 95% CI: 1.10–4.22) (Fig. 2A). Age remained significant in a sensitivity analysis of the same logistic regression model using age as a continuous variable (Supplemental Table 3). In a second sensitivity analysis using a regression model based on age subgroups (Fig. 2B), children aged 3-10 years had a lower risk (aOR: 0.32, 95% CI: 0.10–0.97) and adults ≥80 years of age had a higher risk of detectably expressed ARGs (aOR: 6.21, 95% CI: 1.33–28.99) compared with children aged 0–2 years. The two models in Fig. 2 suggest that HA-LRTI, but not CA-LRTI, may also increase the risk of having detectably expressed ARGs.

In an analysis restricted exclusively to children and accounting for enrollment U.S. census region and the presence of a complex chronic condition, children 3-10 years of age continued to have the lowest risk of having detectably expressed ARGs, however, enrollment site was a significant risk factor (Supplemental Table 4). Given this, we performed a sensitivity analysis limited to pediatric and adult patients from the same U.S. census region (West), and found that the binary age group remained associated with increased risk of ARG detection (unadjusted OR: 2.33, 95% CI: 1.2–4.52).

### Lower Respiratory Tract Microbiome

In the lower respiratory tract of pediatric patients, the total microbiome consisted of a median bacterial proportion of 91.7% (IQR: 11.7–99.8%), median viral proportion of 4.4% (IQR: 0.0–86.9%), and median fungal proportion of 0.0% (IQR: 0.0–0.1%). In adults, the total microbiome consisted of a median bacterial proportion of 99.8% (IQR: 98.3–100.0%), a median viral proportion of 0.0% (IQR: 0.0–0.4%), and a median fungal proportion of 0.0% (IQR: 0.0–0.3%). Available culture and antimicrobial susceptibility testing results are shown in Supplemental Table 5.

The total bacterial abundance of the lung microbiome increased with age (Fig. 3A). Bacterial microbiome alpha diversity initially increased during childhood and adulthood, peaked in the 40–49-year-old age group, and then decreased in older adults (Fig. 3B). The

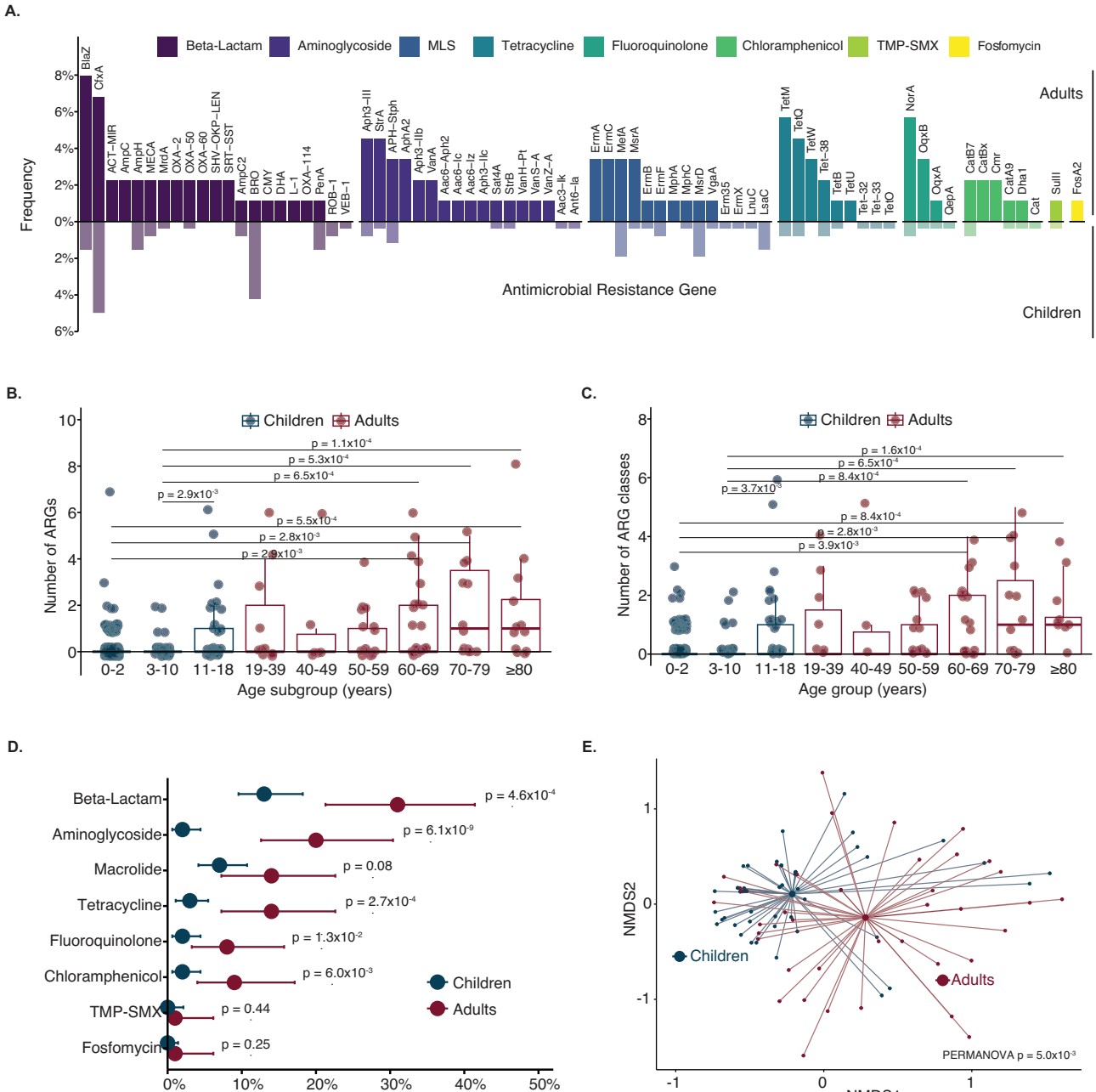

**Fig. 1 | Lung resistome of children compared with adults. A** Frequency of children (translucent) and adults (solid) with each antimicrobial resistance gene (ARG), stratified by ARG class ($n = 349$ patients). Colors indicate the ARG class. **B** Number of ARGs detected in children and adults by age subgroups ($n = 349$ patients). Two outliers were omitted for visualization purposes; one 11–18-year-old patient with 18 ARGs detected and another 70–79 year-old patient with 12 ARGs detected. **C** Number of ARG classes detected in children and adults by age subgroups ($n = 349$ patients). For Figures B and C, $p$ values were calculated using two-sided Wilcoxon-rank sum test and adjusted for multiple comparisons with False Discovery Rate (FDR) correction. Only the statistically significant $p$ values ($p < 0.05$) are depicted. Boxplot elements from Figures B and C include a center line (median), box limits (upper and lower quartiles), and whiskers (1.5x interquartile range). Individual data points are shown using overlaid dot plots. **D** Proportion of patients with ARGs by ARG class, stratified by pediatric and adult cohorts ($n = 349$ patients). The center dots indicate the proportion and the error bars indicate 95% confidence intervals calculated by the Clopper-Pearson exact binomial method. *P* values were obtained by two-sided Pearson's Chi-square test and Fisher's exact test for comparisons with counts of <5 patients. **E** Beta diversity of the antimicrobial resistome of children and adults among those with detectable ARGs. The $p$ value was calculated using the Bray-Curtis dissimilarity index and the PERMANOVA test with 1000 permutations. For Figures B to E, the color indicated children (blue) or adults (red). Source data, including all $p$ values, are provided as a Source Data file. Abbreviation: MLS macrolide-lincosamide-streptogramin, TMP-SMX trimethoprim-sulfamethoxazole, NMDS nonmetric multidimensional scaling.

community composition of the bacterial respiratory microbiome differed between children and adults, based on Bray Curtis dissimilarity index ($p < 0.01$ by PERMANOVA) (Fig. 3C). Differential abundance analysis revealed eight bacterial genera with statistically significant differences in prevalent between children and adults (*Enterococcus*,

*Pseudomonas, Staphylococcus, Bacteroides, Prevotella, Mannheimia, Haemophilus* and *Moraxella*) (Fig. 3D). The most prevalent bacterial species within each of these genera also differed between age groups (Fig. 3E). Four bacterial species were significantly differentially abundant between the age groups (Supplemental Fig. 5).

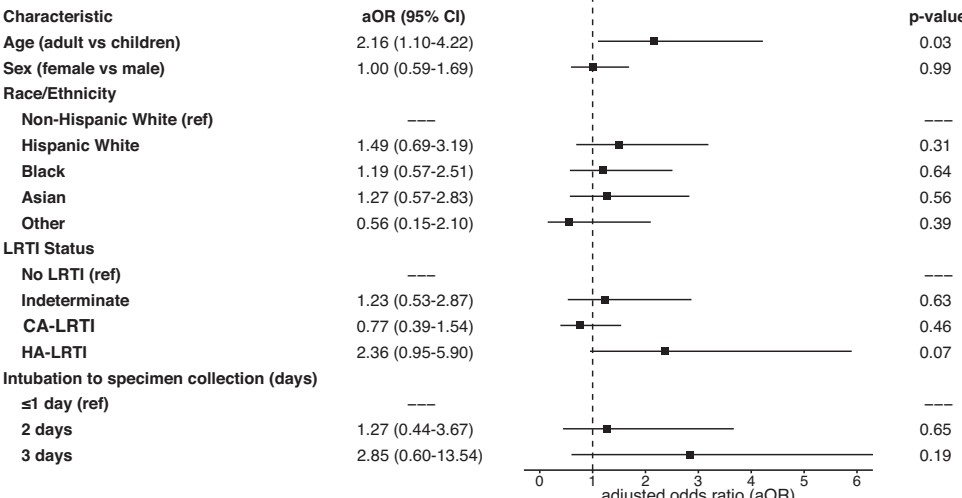

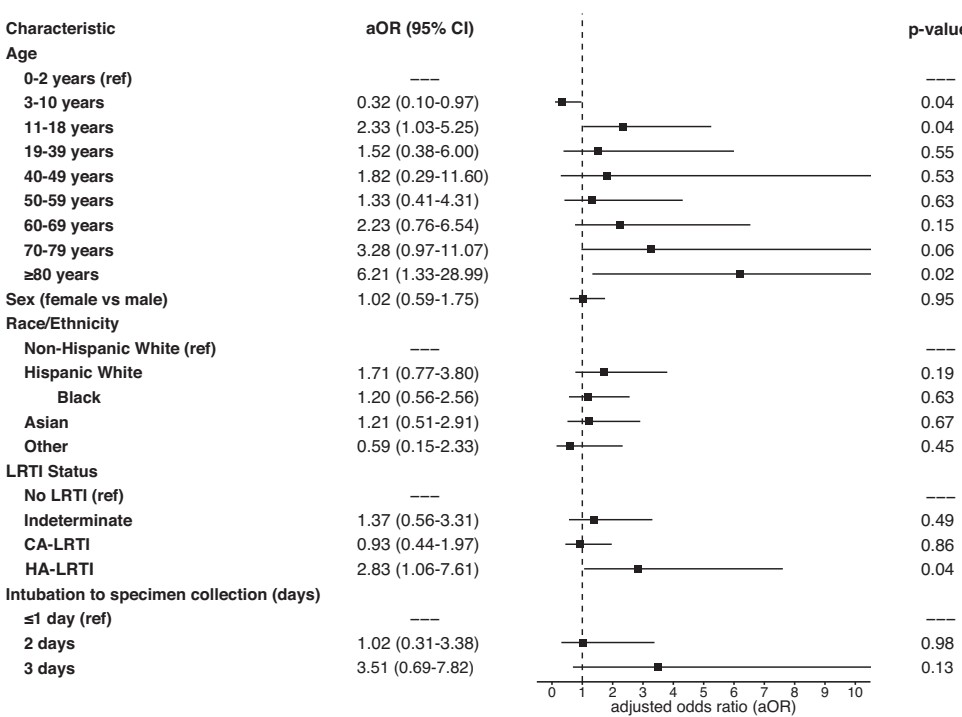

**Fig. 2 | Association of age with presence of antimicrobial resistance genes (ARGs) in the lung resistome.** Multivariable logistic regression model evaluating the association of (**A**) binary age and (**B**) age subgroups with the presence of ARGs, accounting for sex, race/ethnicity, and lower respiratory tract infection (LRTI) status (*n* = 349 patients). The center squares indicate aOR, and the error bars indicate 95% CIs. The *p* value was obtained using a two-sided Wald test. Source data are provided as a Source Data file. Abbreviation: aOR adjusted odds ratio, CI confidence intervals, LRTI lower respiratory tract infection, CA-LRTI community-acquired LRTI, HA-LRTI hospital-acquired LRTI.

## Interactions between the lower respiratory tract microbiome and resistome

In a logistic regression model accounting for total bacterial abundance, bacterial alpha diversity, and LRTI status, the binary age group remained associated with an increased risk of ARG detection (aOR: 2.38, 95% CI: 1.25–4.54) (Fig. 4). Differential abundance analysis revealed seven bacterial genera with statistically significant differences between patients with or without detectable ARGs (Fig. 5). In individual fitted logistic regression models where the outcome was presence of ARG expression, and independent variables included age group, total bacterial abundance, bacterial alpha diversity, LRTI status, and one of the seven differentially abundant bacterial genera, adults remained at increased risk of having detectable ARG expression compared with children, although binary age was no longer statistically significant for models including *Enterococus spp.*, *Pseudomonas spp.*, and *Staphylococcus spp.* (Fig. 4, Supplemental Table 6). In these models, *Fusobacterium spp.* (aOR: 2.18, 95% CI: 1.13–4.22) and *Enterococcus spp.* (aOR: 2.23, 95% CI: 1.08–4.59) were statistically significant risk factors. Sensitivity analyses of all these models using age as a continuous variable found only *Fusobacterium spp.* (aOR: 2.19, 95% CI: 1.13–4.24) as statistically significant (Supplemental Table 7).

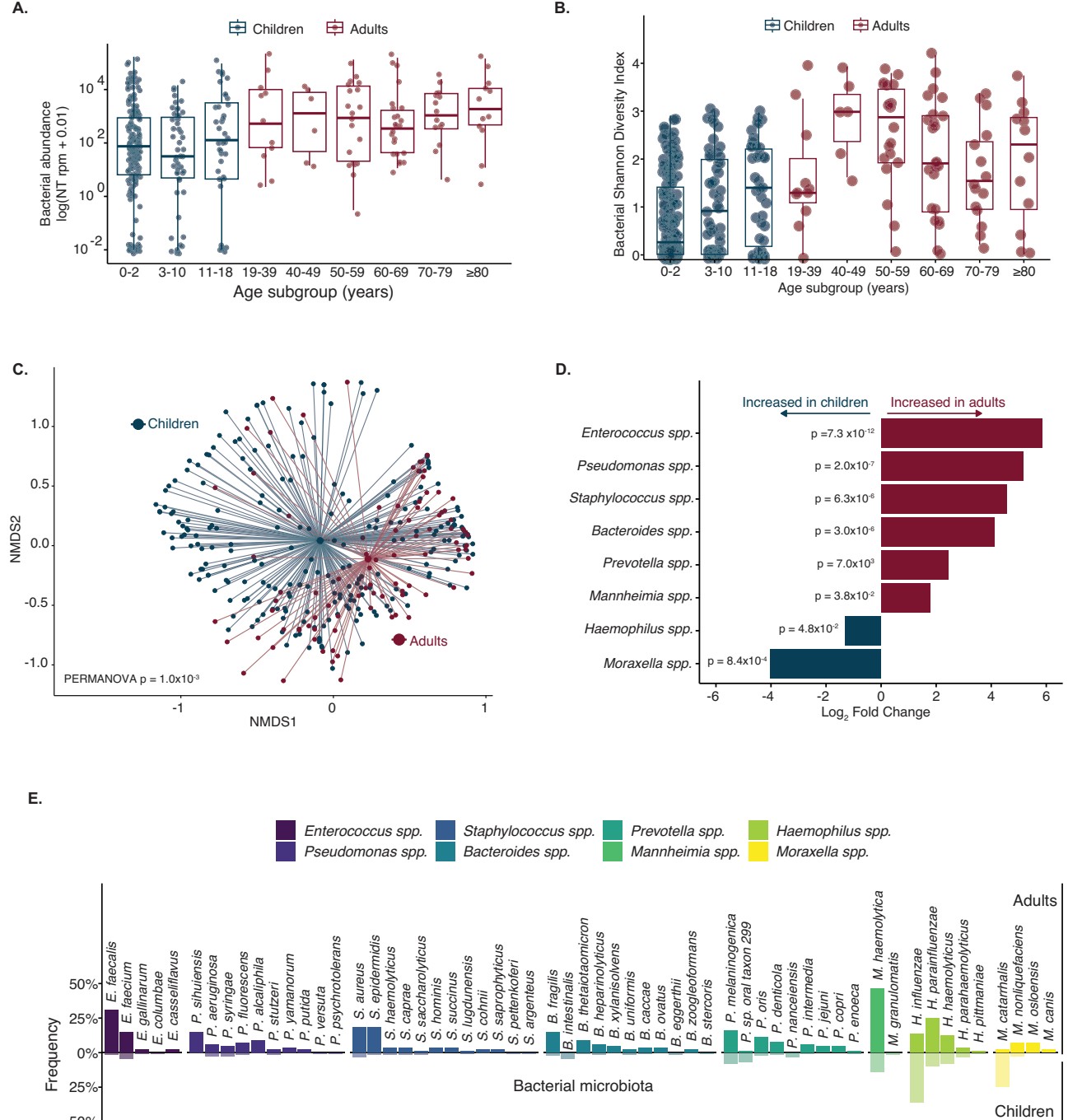

**Fig. 3 | Lung bacterial microbiome of children compared with adults. A** Bacterial abundance in the lung microbiome measured in total bacterial alignments to the NCBI NT database per million reads sequenced (NT rpm) in children and adults by age subgroups ($n = 349$ patients). **B** Alpha diversity, calculated by the Shannon diversity index, of the bacterial lung microbiome of children and adults by age subgroups ($n = 349$ patients). Boxplot elements from Figures A and B include a center line (median), box limits (upper and lower quartiles), and whiskers (1.5x interquartile range). Individual data points are shown using overlaid dot plots. **C** Beta diversity of the bacterial lung microbiome of children and adults. *P* value calculated based on the Bray-Curtis dissimilarity index and the PERMANOVA test with 1000 permutations. For Figures A to C, the color indicated children (blue) or adults (red). **D** Statistically significant (*p* value < 0.05) differential abundant bacterial genera, by $\log_2$ fold change of bacterial counts, detected in children and adults. *P* values were calculated using a two-sided Wald test adjusted for multiple comparisons. Bar colors indicate whether the species was more abundant in children (blue) or adults (red). **E** Frequency of the bacterial species detected in ≥5% of children (translucent) and adults (solid) among the differentially abundant bacterial genera. Colors indicate the bacterial genera. For those with multiple species detected per genus, only the most abundant species was included in this analysis. Source data are provided as a Source Data file. Abbreviations: NT rpm nucleotide reads per million reads sequenced, NMDS nonmetric multidimensional scaling.

In a sub-analysis of 67 patients with no evidence of LRTI, 53 (79%) patients had reads mapping to common bacterial respiratory pathogens detectable at low abundance within the lung microbiome (Supplemental Table 8). Of these patients, 17 (25%) had detectable expression of ARGs, including one patient with an *AmpC* beta-lactamase gene. A similar proportion of patients with and without metatranscriptomic detection of a respiratory pathogen had detectable ARG expression (26% and 21%, respectively; *p* = 0.71). When

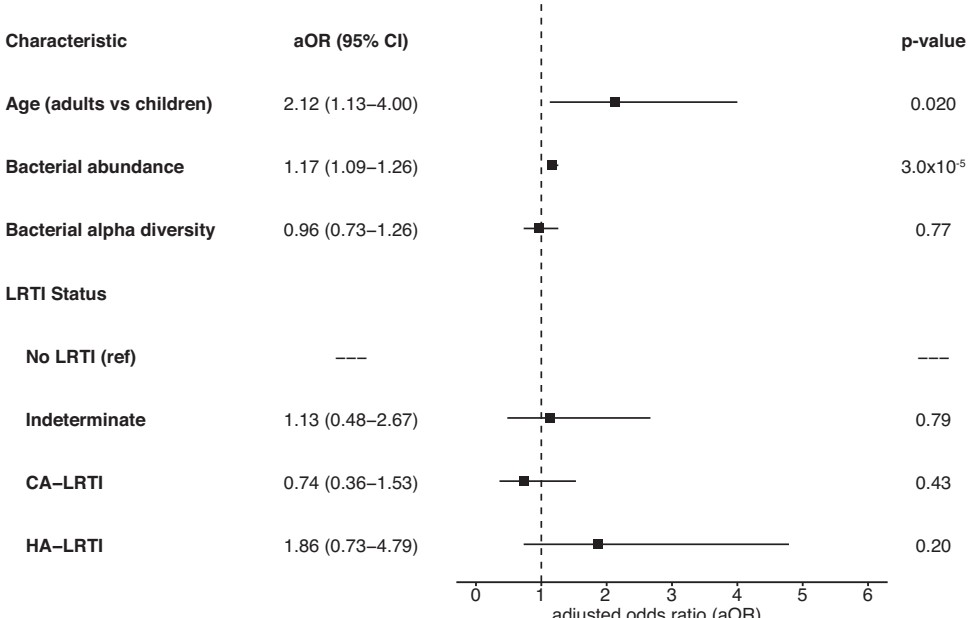

**Fig. 4 | Association of age and the bacterial microbiome with presence of antimicrobial resistance genes (ARGs) in the lung resistome.** Multivariable logistic regression model evaluating the association of binary age with the presence of ARGs, accounting for logarithmic total bacterial abundance [log (NT rpm + 0.01)] per sample, bacterial alpha diversity ($n = 349$ patients). The center squares indicate aOR, and the error bars indicate 95% CIs. The p-value was obtained using a two-sided Wald test. Source data are provided as a Source Data file. Abbreviation: NT rpm nucleotide reads per million, aOR adjusted odds ratio, CI confidence intervals, LRTI lower respiratory tract infection, CA-LRTI community-acquired LRT, HA-LRTI hospital-acquired LRTI.

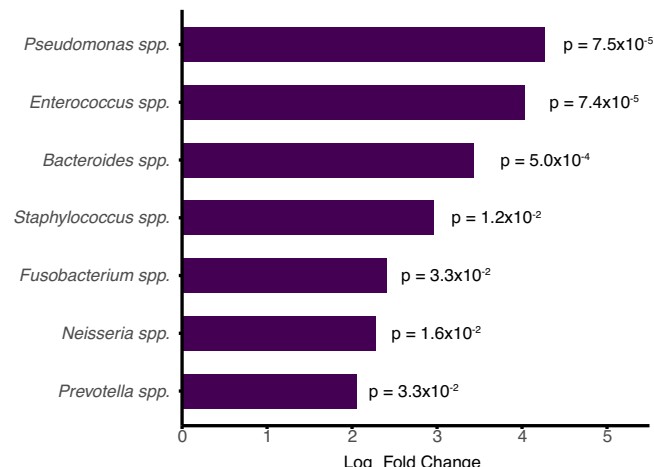

**Fig. 5 | Differentially abundant bacterial genera between patients with antimicrobial resistance genes (ARGs) and patients without ARGs in the lung resistome.** Statistically significant ($p < 0.05$) differentially abundant bacterial genera, by $\log_2$ fold change of bacterial counts, detected in patients with ARGs compared with patients without ARGs. All detected bacterial genera were more prevalent in patients with ARGs compared with patients without ARGs. P values were calculated using a two-sided Wald test adjusted for multiple comparisons. Source data are provided as a Source Data file.

comparing ARG abundance in patients with no LRTI versus patients with LRTI, there was no significant difference ($p = 0.83$) between the two groups (Supplemental Fig. 4).

## Discussion

Utilizing metatranscriptomics, we identify age as an independent risk factor for ARG detection in the lung microbiome among critically ill, recently intubated patients. We find that detection of ARGs in the lower respiratory tract increases across most of the age spectrum, with the oldest patients harboring the highest number of ARGs detectably expressed at the individual gene and class levels. These findings advance our understanding of the lung microbiome as a potential antimicrobial resistance reservoir and highlight its potential contribution to drug-resistant respiratory infections.

Across every ARG class examined, adults had a greater number of ARGs detected compared with children. The majority of detected ARGs in the pediatric cohort conferred resistance to beta-lactams and MLS, while in adults, beta-lactam and aminoglycoside ARGs were most prevalent. Studies of nasal samples from healthy neonates[14] and neonates with cystic fibrosis[15] found a similar expression of beta-lactam ARG dominance, while oral flora samples from children[16] and sputum samples from adults[17] found that MLS resistance genes were most prevalent in the oropharynx. Intriguingly, we found that children 0-2 years of age had a higher proportion of beta-lactam ARGs detected than those 3–10 years of age. This may reflect maternally derived microbial communities and associated ARGs acquired at birth, which go on to comprise the lung microbiome during the first years of life[18].

Our findings differ from gut resistome studies, which have found that tetracycline ARG expression are most prevalent, followed by expression of MLS and beta-lactam ARGs[19]. These differences may reflect differences in the routes of ARG and antibiotic exposure (e.g., inhaled, ingested, intravenous) and highlight potential AMR transmission differences from the lungs compared with the gut. Indeed, recent work demonstrates the presence of ARGs and microbiota in urban air samples[20,21], suggesting that environmental exposures may be a relevant route of lung resistome exchange particularly for patients with prolonged or repeated exposure to healthcare facilities where resistant microbes are more prevalent.

In the U.S., the most prescribed outpatient antibiotics are beta-lactams, fluoroquinolones, and macrolides[22], while beta-lactams, macrolides, and glycopeptides are the most frequently prescribed inpatient antibiotics[23]. While expression of beta-lactam ARGs were the most prevalent ARG class detected in both children and adults, adults had a greater proportion of detectable expression of aminoglycoside and tetracycline ARGs. Given that these antimicrobial classes have

been widely used in the agriculture and livestock industries[24], exposures to environmental bacteria harboring these ARGs over the lifespan could be one possible explanation. Other possible explanations include community exposures to other individuals, co-selection of ARGs on mobile genetic elements carrying multiple ARGs, or cross-resistance due to multi-drug efflux pumps[25].

We observed differences in the lower respiratory tract microbiome with age, including bacterial abundance, diversity, and taxonomic composition. Our findings are in line with a prior study demonstrating that bacterial abundance in the lung microbiome of CF patients increases with age[26]. We also found that bacterial alpha diversity increased in childhood, peaked in middle age, and decreased with older age. The role of endogenous respiratory microbiota in both the pathophysiology and diagnosis of critical illness syndromes in increasingly recognized, and our results suggest that adjusting for age should be considered in clinical and translational studies of the lung microbiome.

Even when accounting for demographic, clinical, and microbiome differences, age remained an independent risk factor for ARG detection. Our findings raise the possibility that selective environmental pressures driving AMR acquisition from the environment may be continuous over the lifespan, shaping the airway microbiome and associated resistome. While the detection of ARG expression in the airway microbiome does not equate to clinically relevant resistance, it suggests the potential for the development of phenotypic resistance[3] with possible implications for patient care. A recent comparison of metatranscriptomics versus phenotypic antibiotic susceptibility testing in the adult cohort demonstrated a sensitivity of 52% and specificity of 86% for gram-positive pathogens, and a sensitivity of 100% and specificity of 64% for gram-negative pathogens[3]. Furthermore, commensal bacteria within the lung microbiome can exchange ARGs via horizontal gene transfer to pathogens or pathobionts, leading to the emergence of drug-resistant LRTI[27–29]. We found detectable ARG expression in 25% of patients without clinical or microbiologic evidence of a lower respiratory tract infection and we observed no difference in ARG abundance between patients with LRTI and patients without LRTI, supporting the idea that commensal microbes or incidentally carried pathogens contribute to the resistome. Further research is needed to better characterize and understand the prevalence, acquisition, and transmission dynamics of ARGs within the lung microbiome.

This study has limitations. First, the patient cohort was demographically and medically complex. While age was found to be a significant factor to the detection of ARG expression, it is possible that some comorbidities, such as chronic lung disease, may also significantly influence the lung resistome. Additionally, there was an uneven distribution of patients across all ages, with a greater number of young children and older adults, reflecting the distribution of the critically ill, mechanically ventilated patient populations. To account for this, we performed sensitivity analyses using age as a continuous variable or using age subgroups. Differences in geographic location and timing of tracheal aspirate collection between age groups was accounted for by incorporating the variables into the multivariable logistic regression models. Second, the study included only patients from the U.S. and may not be representative of the global population. Third, while an equal proportion (90%) of the pediatric and adult cohorts received antibiotics prior to intubation, the specific antibiotic exposures in terms of exact drugs may have differed, affecting ARG expression. It is possible that specific antibiotic exposures may have influenced our metatranscriptomic results. Longitudinal samples to evaluate changes in ARG expression over time, particularly in relation to antibiotic exposure, were not available, and their collection should be prioritized in future studies. Fourth, the detection of ARG expression is biased towards the most abundant taxa in the lung microbiome

and we are likely missing the detection of ARGs from less abundant taxa. Furthermore, without a mass standard, we were only able to calculate bacterial abundance and not absolute bacterial load. Finally, enrollment occurred prior to the COVID-19 pandemic. Thus, ARG abundance and class profiles may be different than in the current population given the increase in antibiotic use and AMR infections since 2020[30,31].

In summary, we demonstrate that age is independently associated with the detection of ARG expression in the lung microbiome in a population of critically ill patients soon after intubation. Our results suggest that healthcare, community, and environmental exposures throughout life may contribute to the reservoir of ARGs in the respiratory tract. Taken together, these findings advance our understanding of AMR in the context of the human microbiome and have implications for the management of infectious diseases, antimicrobial stewardship programs, and public health policies.

## Methods

### Ethics

This study complied with all relevant ethical regulations for work with human participants, and written informed consent was obtained for use of samples and data for this analysis. The pediatric cohort study was approved by a single Institutional Review Board (IRB) at the University of Utah (protocol #00088656). The adult cohort study was approved by the UCSF IRB (protocol #10-02701). For both cohorts, the IRBs approved of an initial waiver consent for obtaining excess respiratory samples and informed consent was subsequently obtained from parents or other legal guardians (pediatric patients) and from patients or their surrogates (adult patients). No compensation was given for study participation.

### Study design and clinical cohorts

We leveraged data from prospective pediatric[32–34] and adult[35] cohorts of patients with acute respiratory failure admitted to intensive care units (ICUs) in the United States (USA). Pediatric patients ($n = 261$), aged 31 days to 18 years, were enrolled from eight tertiary care hospitals in the Collaborative Pediatric Critical Care Research Network (CPCCRN) between February 2015 and December 2017[34,35]. Adults ($n = 88$), aged >18 years, were enrolled from a single tertiary care center in California, USA between July 2013 to October 2017. Exclusion criteria included withdrawal of consent, inability to obtain a TA sample from the patient within 24 hours of intubation (for the pediatric cohort) or 72 hours of intubation (for the adult cohort), any condition for which TA collection was contraindicated, a previous episode of mechanical ventilation during the hospitalization, or previous enrollment in the study. The pediatric cohort had the following additional exclusion criteria: presence of or plans to place a tracheostomy tube, a previous episode of mechanical ventilation during the same hospitalization, family/team lack of commitment to aggressive intensive care as indicated by "do-not-resuscitate" orders and/or other limitation of care. From each patient, TA samples were collected, mixed 1:1 with DNA/RNA shield (Zymo, Inc.), and stored at −80 °C.

Electronic medical records were reviewed to obtain demographics, including age, self-reported sex, and race/ethnicity, and clinical data. LRTI status was retrospectively adjudicated by study physicians based on a previously described algorithm[34,35] grouping patients as follows: 1) LRTI defined clinically, with or without a clinical microbiological diagnosis (LRTI); 2) No clinical or microbiologic evidence of respiratory infection and a clear alternative etiology for the acute respiratory failure (No LRTI); or 3) patients who did not meet either above criteria (Indeterminate). LRTI was further separated into CA-LRTI (LRTI diagnosed within 48 hours of hospital admission), and HA-LRTI (LRTI diagnosed ≥48 hours after hospital admission).

## Metatranscriptomic RNA Sequencing, Taxonomic Alignment, and Detection of ARGs

RNA extracted from tracheal aspirate (TA) specimens underwent library preparation using the NEBNext Ultra II Library Prep Kit (New England BioLabs) and 150 bp paired-end sequencing on an Illumina Novaseq 6000 instrument at University of California San Francisco[34]. Briefly, RNA was extracted from 300 μl of patient TA using bead-based lysis and the Allprep DNA/RNA kit (Qiagen, Inc.), which included a DNase treatment step. RNA was reverse transcribed to generate cDNA, and sequencing library preparation was performed using the NEBNext Ultra II Library Prep Kit (New England Biolabs, Inc.). RNA-Seq libraries underwent paired-end Illumina sequencing on an HiSeq 4000 or Novaseq 6000 instrument.

Quantification of microbial taxa from raw sequencing reads was carried out using the CZ-ID bioinformatics pipeline[36]. The CZ-ID pipeline performs host and quality filtration on the raw mNGS data, then executes an assembly-based alignment against reference microbial genomes from the National Center for Biotechnology Information (NCBI) nucleotide (NT) database[36]. ARGs annotated in the Antibiotic Resistance Gene-ANNOTation (ARG-ANNOT) database[37] were detected using the Short Read Sequence Typing (SRST2) algorithm[38]. Negative control water samples were processed in parallel, and a previously described negative binomial model was used to filter out microbial contaminants from the laboratory environment[34]. ARGs with <5% coverage or found in ≥10% of negative control water samples (*TEM-1D*, *TetC*, *SulI*, *OXA-22*, *Aph3'Ia*, *CatA1*) were excluded from the analysis.

For this analysis, host sequences were removed. In the adult samples, the median proportion of host sequences was 99.66% (IQR: 97.85–99.92%); in the pediatric samples, the median proportion of host sequences was 99.86% (IQR: 99.52–99.96%).

### Statistics and reproducibility

Age was defined in three ways: (1) a binary variable of children (31 days to 18 years) or adults (over 18 years); (2) nine subgroups of 0-2 years, 3–10 years, 11–18 years, 19–39 years, 40–49 years, 50–59 years, 60–69 years, 70–79 years, and ≥80 years; or (3) continuous age in years. We used Pearson's Chi-square test and Fisher's exact test for comparison of categorical demographic and clinical variables. *P* values < 0.05 were considered statistically significant. All analyses were conducted in R (v4.2.1). Figures were made using the following R packages: ggplot2 (v3.4.0), epiDisplay (v 3.5.0.2), and patchwork (v 1.1.2). We analyzed data available from previously described cohorts of critically ill children[32–34] and adults[35]. No data were excluded from analyses and no statistical method was used to predetermine sample size.

### Resistome analyses

The number of ARGs detectably expressed in the lower respiratory tract microbiome of children and adults were compared at the individual gene and ARG class (e.g., beta-lactamase) levels. *P* values were calculated using the two-sided Wilcoxon rank-sum test for nonparametric continuous variables and false discovery rate (FDR) correction was applied for multiple comparisons. We compared the proportion of detected expression of ARG classes by binary age (pediatric versus adult) and by age subgroups. 95% confidence intervals [CI] for population proportions were obtained using the Clopper-Pearson exact binomial method.

ARG abundance was calculated based on the average sequencing read depth across each gene, normalized by gene length and total reads, reported as depth per million (dpm)[38,39]. Resistome alpha diversity was calculated using the Shannon Diversity Index (SDI) and ARG dpm. Beta diversity was calculated on patients with ARGs detected using the Bray-Curtis method with 1000 permutations using the PERMANOVA test and displayed via nonmetric multidimensional scaling (NMDS). Alpha and beta diversity calculations were performed using the R package *vegan* (v2.6.4)[40].

A multivariable logistic regression model incorporating demographic and clinical characteristics (sex, race/ethnicity, LRTI status, days from intubation to specimen collection) was used to determine associations between binary age (adults vs children) and detection of ARG expression. Additional regression models were performed using: (1) age years as a continuous variable, and (2) the previously defined nine age subgroups. To assess for potential geographic differences in ARG expression, an additional analysis was performed within the pediatric cohort only and included adjustment for U.S. census region and presence of complex chronic conditions; the latter was defined by a previously validated pediatric medical complexity algorithm[41]. A sensitivity analysis limited to pediatric and adult patients from the same U.S. census region was also performed. 95% confidence intervals (CI) for the multivariable logistic regression models were calculated using the Wald CI. Observations with missing data are excluded.

### Microbiome analyses

We describe the proportion of virus, bacterial, and fungal abundance in the respiratory tract microbiome and the available culture and antimicrobial susceptibility testing results of the pediatric and adult patients. For the remainder of the microbiome and ARG analyses, we focused on the bacterial microbiome. We assessed the respiratory tract bacterial microbiome of children and adults to evaluate age-related differences in taxonomic composition and diversity, which we considered as possible confounders or mediators of the relationship between age and detectably expressed ARGs. We assessed microbiota at the genus level, calculated total bacterial abundance (measured in reads per million, rpM), and calculated bacterial alpha diversity across age subgroups. We further stratified by LRTI status (CA-LRTI, HA-LRTI, No LRTI). Lung microbiome beta diversity calculations were carried out using the Bray-Curtis dissimilarity index and PERMANOVA to assess statistical significance. Differential abundance analysis was performed using the R package *DESeq2* (v1.36.0)[42] by assessing bacterial genera and taxa in the lung microbiome present in ≥20% of patients. We also described the prevalence of the most abundant species within each differentially expressed genus.

### Associations between the microbiome and resistome analyses

To test whether age-related differences in the lung microbiome might influence ARG results, we carried out additional analyses adjusting for bacterial abundance and alpha diversity. To test whether specific taxa might influence age-related changes in ARG expression, we performed a differential abundance analysis of bacterial genera detected in patients with or without detectable expression of ARGs, using *DESeq2*[42]. Subsequently, for each differentially abundant genus, we fit individual regression models for the outcome of having ARGs detected, accounting for binary age, bacterial abundance, alpha diversity, LRTI status, and presence of one of the differentially abundant genera. Additional sensitivity analyses were performed for these models using age as a continuous variable. Lastly, to evaluate the potential role of commensal organisms on the resistome, we performed (1) a sub-analysis of detectable ARG expression in patients with a no LRTI diagnosis, and (2) compared ARG abundance between patients with LRTI and patients with no LRTI.

### Reporting summary

Further information on research design is available in the Nature Portfolio Reporting Summary linked to this article.

## Data availability

FASTQ files containing the raw microbial reads used for the metatranscriptomic analyses in this study have been deposited in the NCBI Sequence Read Archive (SRA) database under BioProject accessions PRJNA875913 for the pediatric cohort and PRJNA450137 for the adult cohort. The NCBI NT database is available at: ftp://ftp.ncbi.nlm.nih.

gov/blast/db/FASTA/. The Antibiotic Resistance Gene-ANNOTation (ARG-ANNOT) database is available at: https://github.com/katholt/srst2/tree/master/data. The processed data are available at the Github repository: https://github.com/victoriatchu/agingAMR [https://doi.org/10.5281/zenodo.10258295][43] for reviewing and reproducing the analyses. The source data generated in this study for the figures are provided in the Source Data file with this paper. Source data are provided with this paper.

## Code availability

The code is available at: https://github.com/victoriatchu/agingAMR [https://doi.org/10.5281/zenodo.10258295][43].

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

## Acknowledgements

This work was supported in part by the following grants from NHLBI: R01HL155418 (CRL, PMM), R01HL124103 (PMM), K23HL138461 (CRL), and R53HL140026 (CSC), as well as grants from the Chan Zuckerberg Bio-hub (VTC, CRL, JLD). The investigators thank all patients and their families for participating in this project. We also would like to acknowledge the contributions from principal investigators, co-investigators, research coordinators, and allied research personnel at the following sites: University of California San Francisco, San Francisco, CA; Children's Hospital of Colorado, Aurora, CO; Chan Zuckerberg Biohub; Children's Hospital of Michigan, Detroit, MI; Children's Hospital of Philadelphia, Philadelphia, PA; Children's National Medical Center, Washington, DC; Nationwide Children's Hospital, Columbus, OH; Mattel Children's Hospital, University of California Los Angeles, Los Angeles, CA; Children's Hospital of Pittsburgh, University of Pittsburgh Medical Center, Pittsburgh, PA; University of Utah; and Data Coordinating Center, Salt Lake City, Utah.

## Author contributions

V.T.C., L.A., P.M.M., C.R.L. were involved in the conceptualization. M.A.M., C.S.C., and P.M.M. were involved in the cohort design. A.T., E.M., L.A., C.M.O., and B.D.W. were involved in the data curation, and A.T., E.M., and C.M.O. were involved in the data management. V.T.C., K.L.K., A.G., J.L.D., C.R.L. were involved in the methodology. K.L.K. was involved with the software. V.T.C. analyzed the data. V.T.C. and C.R.L. wrote the original draft. L.A., M.A.M., J.L.D., C.S.C., P.M.M., and C.R.L. were involved in the funding acquisition. C.R.L. was also involved in the project administration, obtaining resources, providing supervision, and validation. All authors were involved in the reviewing and editing of the manuscript.

## Competing interests

The authors declare no competing interest.
