## [Peer Review File · Nature Communications]

REVIEWER COMMENTS

Reviewer #1 (Remarks to the Author):

In this study, Chu et al., used tracheal aspirates of children and adults to perform RNA sequencing to look for antimicrobial resistance gene expression and composition of the airway microbiome. 261 pediatric cases came from 8 centers while 88 adults came from a single center. Older adults had greater antimicrobial resistance gene expression

Overall, this is an elegant study, multicenter, involving multiple subjects and using sequencing data to focus on the resistome, which is commonly overlooked. My comments are related to some analytical concerns. The idea that age and time to be exposed to antibiotics is associated with development of antibiotic resistance might not be that novel but showing it on a large cohort with lower airway samples is potentially quite impactful and should be of interests to readers..

Below are my point-by-point comments

Major Comments

1. What was the depth of the sequence data? How many reads were annotated to resistant genes per sample. Was the data rarefied or transformed to relative abundance for beta diversity analyses presented in Figure 1E.
2. The authors describe the composition of the bacterial microbiome based on their metatranscriptome data but there is no mention to the non bacterial fraction, fungi, viruses.
3. Line 247: "Differential abundance analysis revealed eight bacterial genera with statistically significant differences in abundance between children and adults (Enterococcus, Pseudomonas, Staphylococcus, Bacteroides, Prevotella, Mannheimia, Haemophilus and Moraxella). The most abundant bacterial species within each of these genera also differed between age groups (Figure 3E)". However the only data shown is differences in prevalence. Were the relative abundances also different? Where is that data?
4. To calculate the "total bacterial abundance", they use total bacterial alignments to the NCBI NT database per million reads sequenced. While this might be OK, some of the differences might be related to how much mammalian RNA there is in the data, which might be influenced by how much inflammation there is among patients. Thus this total bacterial abundance might not be a good representation of the bacterial load. I'm not saying that this is wrong but I would be cautious not to confuse a naïve reader into thinking that this is equivalent to bacterial load.
5. Figure 4 B tries to address which taxa might be contributing to ARG by comparing taxonomic abundance among those with vs. those without ARGs. Alternatively, if there is sufficient depth and

coverage, one could try to build contigs including ARG and evaluating their taxonomic origin. I understand that this may not be possible for most ARGs since it requires quite a bit of good coverage but for those where the relative abundance is high concordantly with high abundance of certain taxa, it might be possible. This might be quite interesting since the question is whether only taxa commonly considered “classical respiratory pathogens”, such as *P aeruginosa* or *S aureus*, are the main source of ARGs, or who else contributes to these ARGs, which would actually be even more interesting...

6. Another way to evaluate for the above would be to look for presence of ARG among samples where there was culture negativity for respiratory pathogens and also no detectable reads from most common respiratory pathogens. This goes back to what they discuss later: “Commensal bacteria within the lung microbiome can exchange ARGs via horizontal gene transfer to pathogens or pathobionts, leading to the emergence of drug resistant LRTI”

7. There is no comparison with culture results, including antimicrobial resistance, which I would expect to be available for at least a proportion of the patients.

8. The authors have measured presence of transcripts for ARG, not presence of genes (which could have been done by using WGS). Thus, they measured gene expression and it is possible that the genomic potential is different. The authors should revise several sections where they talk about detecting ARGs clarifying that they are detecting expression of those genes or their transcripts.

9. In line with the above, it would have been great to have a set of longitudinal samples obtained prior and post antibiotic use to evaluate for changes in ARG expressions. The trends seen in the modeling presented in Figure 2 associating higher prevalence of ARG as samples are obtained after more days of being intubated would be supportive of the effects of antibiotic pressure. This may not be available at this point but might be worth discussing.

Minor Comments

1. Line 30: “ranging in age from 31 days to ≥ 89 years”. Age should not state to “equal or greater than” Just state the range

2. Line 221: “ARG alpha diversity as measured by the Shannon Diversity Index increased primarily in patients ≥ 60 years of age (Figure 1D).” That’s in supplementary figure 1B

3. Figure 4 panel B, please indicate if the “Statistically significant ($p < 0.05$) differentially abundant bacterial genera” is adjusted by multiple comparisons.

Reviewer #2 (Remarks to the Author):

Chu et al. describe a metatranscriptomic analysis of two cohorts of patients ranging in age from infant (<1 yo) to elderly adult. Samples were acquired tracheal aspirates of patients requiring mechanical ventilation and were obtained within <72 hours of intubation. The analysis of their data demonstrate that that expressed ARGs are increased with age, with adults typically having a greater number and diversity of ARGs expressed compared to children. Compositional analysis of LRT microbiome showed that bacterial abundance tended to be increased in adults with different bacteria dominating the adult airways compared to children (eg Haemophilus and Moraxella). The paper was generally well written and the figures were clear but interpretation of their findings was complicated by the complex cohort and sequencing approach. Additionally, there is a concern that the cohort may not be well suited to make generalizations about the lung microbiome in otherwise healthy individuals.

Specific major concerns are as follows:

- The significance and implications of using a metatranscriptomic sequencing for this study is not clearly introduced or discussed. This approach seems novel for respiratory resistome research, but there needs to be a clear description of the pros and cons of this approach for resistome studies. For instance:
 - o Many resistome studies are performed using whole metagenomic (DNA) sequencing, which would presumably be more sensitive to detecting ARGs compared to metatranscriptomic sequencing. There are probably many unique advantages to looking at RNA, but they are not mentioned.
 - o Some ARGs are likely induced by antibiotic exposure so looking at expression alone would potentially bias toward detection of ARGs reflecting recent antibiotic exposure. Given that adults are probably more likely to receive a greater variety of antibiotics, this introduces some uncertainty into what their key findings imply. In other words, couldn't the increased transcription of ARG genes reflect recent exposure to more classes of antibiotics in adults? (and not due to differences in ARG gene content).
- The cohort of patients is demographically and medically complex. While this does potentially increase the significance of their findings, it also presents complications in interpretation.
 - o As currently presented, the manuscripts conclusions seem too broad, eg the title "The antibiotic resistance reservoir of the lung microbiome expands with age". The reported findings only apply to the population of patients requiring mechanical ventilation, who are more likely to have comorbid conditions that put them in frequent contact with healthcare settings. It seems possible that some comorbidities would have a greater effect on the lung resistome than age (eg those with chronic lung diseases) and this possibility is not addressed in their analysis.
 - o Most of the patients had a LRTI, or were suspected to have a LRTI. The causal pathogen (if one is defined) was not defined here and presents problems with interpretation. One explanation of their data is that the increase in ARGs with age reflects age-related differences in bacterial etiologies of LRTI, which would be expected over this range of ages.

o It is unclear why LRTI, possible LRTI, and no LRTI are treated as one group since they would be expected to be very different. I.e., defining the lung as a reservoir of ARG in the setting of an active infection seems problematic if the ARGs reside in the causal pathogen—which would imply the reservoir is actually where ever the pathogen originated from (environment, upper respiratory tract, GI, etc).

o Most patients received antibiotics prior to sample collection, which presumably would affect ARG expression data. If available, the potential interaction of antibiotic exposure and ARG expression should be examined.

Minor comments:

- Fig 4; not sure how “bacterial abundance” has p-value of <0.01 when the aOR=1.
- The cross-sectionality of the cohort and its limitations needs to be addressed.

Reviewer #1 (Remarks to the Author):

In this study, Chu et al., used tracheal aspirates of children and adults to perform RNA sequencing to look for antimicrobial resistance gene expression and composition of the airway microbiome. 261 pediatric cases came from 8 centers while 88 adults came from a single center. Older adults had greater antimicrobial resistance gene expression.

Overall, this is an elegant study, multicenter, involving multiple subjects and using sequencing data to focus on the resistome, which is commonly overlooked. My comments are related to some analytical concerns. The idea that age and time to be exposed to antibiotics is associated with development of antibiotic resistance might not be that novel but showing it on a large cohort with lower airway samples is potentially quite impactful and should be of interests to readers..

Major Comments:

1. What was the depth of the sequence data? How many reads were annotated to resistant genes per sample. Was the data rarefied or transformed to relative abundance for beta diversity analyses presented in Figure 1E.

We added several sentences to the end of Paragraph 1 in the Results section to clarify (line 222): "Median total reads per sample for adults was 57.55 million reads (IQR: 47.16-78.19 million reads); medial total reads per sample for children was 80.53 million reads (IQR: 54.09-123.95 million reads). Reads and average read depth, normalized for ARG length, per ARG were a median of 4.00 reads (IQR: 2.00-8.25 reads) and 1.00x read depth (IQR: 0.99-1.98x read depth), respectively, for adults, and a median of 4.00 reads (IQR: 2.00-14.50 reads) and 1.63x read depth (IQR: 0.99-2.94x read depth), respectively, for children." Data was not rarefied or transformed to relative abundance for beta diversity analyses as DESeq2 (the R package used for the differential abundance analysis) relies on count data.

2. The authors describe the composition of the bacterial microbiome based on their metatranscriptome data but there is no mention to the non bacterial fraction, fungi, viruses.

Thank you for this comment. We have added this information to the results (line 277): "In the lower respiratory tract of pediatric patients, the total microbiome consisted of a median bacterial proportion of 91.7% (IQR: 11.7-99.8%), median viral proportion of 4.4% (IQR: 0.0-86.9%), and median fungal proportion of 0.0% (IQR: 0.0-0.1%). In adults, the total microbiome consisted of a median bacterial proportion of 99.8% (IQR: 98.3-100.0%), median viral proportion of 0.0% (IQR: 0.0-0.4%), and median fungal proportion of 0.0% (IQR: 0.0-0.3%)."

3. Line 288: "Differential abundance analysis revealed eight bacterial genera with statistically significant differences in abundance between children and adults (Enterococcus, Pseudomonas, Staphylococcus, Bacteroides, Prevotella, Mannheimia, Haemophilus and Moraxella). The most abundant bacterial species within each of these genera also differed between age groups (Figure 3E)". However the only data shown is differences in prevalence. Were the relative abundances also different? Where is that data?

We thank the reviewer for raising this point. We have changed the sentence on line 291 to say, "The most prevalent bacterial species within each of these genera also differed between age groups (Figure 3E)." We have also added a new Supplemental Figure 5 and the following sentence on line 294, "Four bacterial species were significantly differentially abundant between the age groups (Supplemental Figure 5)."

4. To calculate the “total bacterial abundance”, they use total bacterial alignments to the NCBI NT database per million reads sequenced. While this might be OK, some of the differences might be related to how much mammalian RNA there is in the data, which might be influenced by how much inflammation there is among patients. Thus this total bacterial abundance might not be a good representation of the bacterial load. I’m not saying that this is wrong but I would be cautious not to confuse a naïve reader into thinking that this is equivalent to bacterial load.

We appreciate the reviewer highlighting this point and have now added data on the fraction of host sequences in samples from each cohort with the following sentence in the Methods section (line 130): “For this analysis, host sequences were removed. In the adult samples, the median proportion of host sequences was 99.66% (IQR: 97.85-99.92%); in the pediatric samples, the median proportion of host sequences was 99.86% (IQR: 99.52-99.96%).”

We have also clarified that bacterial abundance differs from total bacterial load with the following sentence in the discussion on line 396: “Furthermore, without a mass standard, we were only able to calculate bacterial abundance and not absolute bacterial load.” Doing so will be a priority in future studies validating these findings.

5. Figure 4 B tries to address which taxa might be contributing to ARG by comparing taxonomic abundance among those with vs. those without ARGs. Alternatively, if there is sufficient depth and coverage, one could try to build contigs including ARG and evaluating their taxonomic origin. I understand that this may not be possible for most ARGs since it requires quite a bit of good coverage but for those where the relative abundance is high concordantly with high abundance of certain taxa, it might be possible. This might be quite interesting since the question is whether only taxa commonly considered “classical respiratory pathogens”, such as *P aeruginosa* or *S aureus*, are the main source of ARGs, or who else contributes to these ARGs, which would actually be even more interesting...

Thank you for this comment. We agree that is a very important and interesting question and appreciate the reviewer’s suggestions for alternative approaches to determine bacterial taxa associated with ARGs. Per the reasons already stated by the reviewer, ARG origin is difficult to evaluate using short read sequencing. However, we took the approach suggested in Comment #6 to address this.

6. Another way to evaluate for the above would be to look for presence of ARG among samples where there was culture negativity for respiratory pathogens and also no detectable reads from most common respiratory pathogens. This goes back to what they discuss later: “Commensal bacteria within the lung microbiome can exchange ARGs via horizontal gene transfer to pathogens or pathobionts, leading to the emergence of drug resistant LRTI”.

Thank you for this suggestion. We approached this in two ways:

First, we compared the abundance of ARGs in patients with LRTI versus those with no clinical evidence of LRTI (culture negative, no clinical evidence of respiratory infection, and an alternative explanation for their respiratory compromise), and found no difference between groups. This highlights the contribution of commensal organisms, or incidentally carried pathogens, to the respiratory resistome, and we have added this analysis in a new Supplemental Figure 4.

Supplemental Figure 4. Antimicrobial resistance gene (ARG) abundance normalized by gene length (average ARG sequencing depth per million reads sequenced, dpm) of patients with lower respiratory tract infection (LRTI) compared with patients no LRTI. P-value was calculated using Wilcoxon-rank sum test.

In addition, we carried out a sub-analysis limited to patients with no evidence of LRTI and found that 25% of these patients still had detectable expression of ARGs. As suggested by the reviewer, we looked at the proportion of patients with metagenomic detection of common bacterial respiratory pathogens and noted that the majority of the “no LRTI” group had reads mapping to respiratory pathogens in their lung microbiome (albeit at low abundance), presumably representing incidental pathogen carriage. We have added a new Supplemental Table 8 to highlight the proportion of no LRTI patients who had both reads mapping to established respiratory pathogens and detectable ARG expression.

We have also added the following text to the results section (line 312): “In a sub-analysis of 67 patients with no evidence of LRTI, 53 (79%) patients had reads mapping to common bacterial respiratory pathogens detectable at low abundance within the lung microbiome (Supplemental Table 8). Of these patients, 17 (25%) had detectable expression of ARGs, including one patient with an AmpC beta-lactamase gene. A similar proportion of patients with and without metatranscriptomic detection of a respiratory pathogen had detectable ARG expression (26% and 21%, respectively; p-value: 0.71). When comparing ARG abundance in patients with no LRTI versus patients with LRTI, there was no significant difference ($p = 0.83$) between the two groups (Supplemental Figure 4).”

We have also added to our discussion the following (line 375): “We found detectable ARG expression in 25% of patients without clinical or microbiologic evidence of a lower respiratory tract infection, and we observed no difference in ARG abundance between patients with LRTI and patients without LRTI, supporting the idea that commensal microbes or incidentally carried pathogens contribute to the resistome.”

7. There is no comparison with culture results, including antimicrobial resistance, which I would expect to be available for at least a proportion of the patients.

We appreciate this point and have added the available culture and susceptibility results as a new Supplemental Table 5. Given prior work evaluating genotype/phenotype relationships between ARGs and phenotypic antimicrobial resistance in the adult cohort (Serpa et al. Genome Medicine, 2022) we did not include these analyses in this manuscript. We now reference this work, however, in the discussion (line 370): “A recent comparison of metatranscriptomics versus phenotypic antibiotic susceptibility testing in the adult cohort demonstrated a sensitivity of 52% and specificity of 86% for gram-positive pathogens, and a sensitivity of 100% and specificity of 64% for gram-negative pathogens.”

8. The authors have measured presence of transcripts for ARG, not presence of genes (which could have been done by using WGS). Thus, they measured gene expression and it is possible that the genomic potential is different. The authors should revise several sections where they talk about detecting ARGs clarifying that they are detecting expression of those genes or their transcripts.

Thank you for raising this important point. We have gone through all references to ARGs in the paper and have clarified that we are referring to detection of ARG expression.

9. In line with the above, it would have been great to have a set of longitudinal samples obtained prior and post antibiotic use to evaluate for changes in ARG expressions. The trends seen in the modeling presented in Figure 2 associating higher prevalence of ARG as samples are obtained after more days of being intubated would be supportive of the effects of antibiotic pressure. This may not be available at this point but might be worth discussing.

We thank the reviewer for this comment, and agree that longitudinal sampling would be ideal for evaluating changes in ARG expression with respect to antibiotic exposure. Unfortunately, longitudinal samples were not available. We recognize the importance of the reviewer’s comments, and have ongoing work in a new longitudinal cohort to evaluate this in more depth. We have added the following line to the limitations section in the discussion to address this point (line 393), “Longitudinal samples to evaluate changes in ARG expression over time, particularly in relation to antibiotic exposure, were not available.”

Minor comments:

1. Line 30: “ranging in age from 31 days to ≥ 89 years”. Age should not state to “equal or greater than”.... Just state the range.

Due to HIPPA protected health information requirements restricting the reporting of age 90 years or older, which is considered identifiable information, we are required to report the age range as stated.

2. Line 221: “ARG alpha diversity as measured by the Shannon Diversity Index increased primarily in patients ≥ 60 years of age (Figure 1D).” That’s in supplementary figure 1B.

Thank you for this comment. We have reviewed the sentence and the figures, and ensured that they are now correctly labeled and referring to the correct figures.

3. Figure 4 panel B, please indicate if the “Statistically significant ($p < 0.05$) differentially abundant bacterial genera” is adjusted by multiple comparisons.

We have added the clause, “adjusted for multiple comparisons,” in the Figure 4B legend.

Reviewer #2 (Remarks to the Author):

Chu et al. describe a metatranscriptomic analysis of two cohorts of patients ranging in age from infant (<1 yo) to elderly adult. Samples were acquired tracheal aspirates of patients requiring mechanical ventilation and were obtained within <72 hours of intubation. The analysis of their data demonstrate that that expressed ARGs are increased with age, with adults typically having a greater number and diversity of ARGs expressed compared to children. Compositional analysis of LRT microbiome showed that bacterial abundance tended to be increased in adults with different bacteria dominating the adult airways compared to children (eg Haemophilus and Moraxella). The paper was generally well written and the figures were clear but interpretation of their findings was complicated by the complex cohort and sequencing approach. Additionally, there is a concern that the cohort may not be well suited to make generalizations about the lung microbiome in otherwise healthy individuals.

1. The significance and implications of using a metatranscriptomic sequencing for this study is not clearly introduced or discussed. This approach seems novel for respiratory resistome research, but there needs to be a clear description of the pros and cons of this approach for resistome studies. For instance: Many resistome studies are performed using whole metagenomic (DNA) sequencing, which would presumably be more sensitive to detecting ARGs compared to metatranscriptomic sequencing. There are probably many unique advantages to looking at RNA, but they are not mentioned.

Thank you for this comment. We have added a sentence in the introduction to further explain our rationale, which includes prior findings from our group demonstrating that RNA-seq may have higher sensitivity and comparable specificity versus DNA-seq for genomic prediction of phenotypic resistance in patients with drug resistant bacterial pneumonia (line 78): “Prior work suggests that RNA-seq may have higher sensitivity and specificity for ARG detection compared with DNA metagenomic sequencing³. The requirement for sample collection into specialized nucleic acid stabilization reagents, RNase-free laboratory equipment, and ultra-cold storage, however, can make RNA-seq more costly and complex for the analysis of clinical samples.”

2. Some ARGs are likely induced by antibiotic exposure so looking at expression alone would potentially bias toward detection of ARGs reflecting recent antibiotic exposure. Given that adults are probably more likely to receive a greater variety of antibiotics, this introduces some uncertainty into what their key findings imply. In other words, couldn't the increased transcription of ARG genes reflect recent exposure to more classes of antibiotics in adults? (and not due to differences in ARG gene content).

We appreciate this comment and would like to note that 90% of patients from both cohorts (pediatric and adult) received antibiotics prior to intubation (Supplementary Table 1). Given this high percentage and given that we only have yes/no data regarding receipt of antibiotics, we are unable to evaluate whether the increased transcription of ARG genes is due to exposure to more classes of antibiotics in adults. We hope to

address this in future studies—currently, we are analyzing a different cohort with longitudinal sampling and more detailed antibiotic use data to better answer this question. However, we believe that the finding that adults have a greater number of detectably expressed ARGs compared to children at the time of intubation is still an important finding regardless of causal factors (although this is an important next step).

3. The cohort of patients is demographically and medically complex. While this does potentially increase the significance of their findings, it also presents complications in interpretation. As currently presented, the manuscripts conclusions seem too broad, eg the title “The antibiotic resistance reservoir of the lung microbiome expands with age”. The reported findings only apply to the population of patients requiring mechanical ventilation, who are more likely to have comorbid conditions that put them in frequent contact with healthcare settings. It seems possible that some comorbidities would have a greater effect on the lung resistome than age (eg those with chronic lung diseases) and this possibility is not addressed in their analysis.

Thank you for this comment. We have changed the title to “The antibiotic resistance reservoir of the lung microbiome expands with age in a population of critically ill, mechanically ventilated patients”.

Furthermore, we have highlighted that this study is limited to critically ill, mechanically ventilated patients in the conclusion of the abstract (line 52): “Taken together, our results demonstrate that age is an independent risk factor for detection of ARG expression in the lower respiratory tract microbiome of critically ill, mechanically ventilated patients.” We have also made clear in the discussion section that we are discussing the critically ill, mechanically ventilated population, and added the following sentence to limitations (line 381): “First, the patient cohort was demographically and medically complex. While age was found to be a significant factor to detection of ARG expression, it is possible that some comorbidities, such as chronic lung disease, may also significantly influence the lung resistome.”

4. Most of the patients had a LRTI, or were suspected to have a LRTI. The causal pathogen (if one is defined) was not defined here and presents problems with interpretation. One explanation of their data is that the increase in ARGs with age reflects age-related differences in bacterial etiologies of LRTI, which would be expected over this range of ages.

Thank you for this comment. We also expected a priori to observe differences between the different LRTI groups, and thus to specifically address this possibility, we added LRTI status as a covariate in both the primary multivariable logistic regression models, and in the additional sensitivity models used in our study.

To further address this point, we have also now added a new Supplemental Table 5 with details on the causative bacterial LRTI pathogens recovered in culture. We note that prevalence of bacterial LRTI pathogens does differ between age cohorts, largely reflecting the significantly higher proportion of adults with hospital acquired (HA)-LRTI compared with children (Supplemental Table 1). With respect to this, we note that we identified HA-LRTI, but not community acquired (CA)-LRTI as a probable risk factor for detection of expressed ARGs. This is highlighted in Figure 2a, which demonstrates that HA-LRTI is associated with an aOR of 2.36 (though the wide confidence interval crosses 1, reflecting the small numbers in this group). To further address the potential contribution of HA vs CA LRTI pathogens to our findings, we have now included CA-

LRTI and HA-LRTI as covariates in the regression models. Importantly, even when accounting for the increased independent risk associated with HA-LRTI (and the related differences in causative pathogens in HA-LRTI patients), age still remained significant.

We have now added a line to better address this point (line 264): “The two models in Figure 2 suggest that HA-LRTI, but not CA-LRTI, may also increase the risk of having detectably expressed ARGs.”

5. It is unclear why LRTI, possible LRTI, and no LRTI are treated as one groups since they would be expected to be very different. I.e, defining the lung as a reservoir of ARG in the setting of an active infection seems problematic if the ARGs reside in the causal pathogen—which would imply the reservoir is actually where ever the pathogen originated from (environment, upper respiratory tract, GI, etc).

We thank the reviewer for this comment. We agree that LRTI status plays a significant role and is an important risk factor to take into consideration. As noted above, because of this, we included it as a covariate in all our multivariable regression models.

In addition, we have added a new analysis comparing the abundance of ARGs in patients with LRTI versus those with no clinical or microbiologic evidence of LRTI, which demonstrated no difference between groups. We believe this highlights the contribution of commensal organisms to the respiratory resistome, and these results are presented in a new Supplemental Figure 4.

Supplemental Figure 4. Antimicrobial resistance gene (ARG) abundance normalized by gene length (average ARG sequencing depth per million reads sequenced, dpm) of patients with lower respiratory tract infection (LRTI) compared with patients no LRTI. P-value was calculated using Wilcoxon-rank sum test.

We would also like to note that we found detectably expressed ARGs in 25% of patients in the “no LRTI” group. While without question bacterial LRTI pathogens would be expected to contribute significantly to the lung resistome, our findings suggest that the microbiome in patients without LRTI also significantly contributes to the resistome. Kindly refer to response to Reviewer #1, Comment #6 for additional discussion of this topic.

We have added the following text to the results section (line 317): “When comparing ARG abundance in patients with no LRTI versus patients with LRTI, there was no significant difference ($p = 0.83$) between the two groups (Supplemental Figure 4).”

We have also added to our discussion the following (line 375): “We found detectable ARG expression in 25% of patients without clinical or microbiologic evidence of a lower respiratory tract infection, and we observed no difference in ARG abundance between patients with LRTI and patients without LRTI, supporting the idea that commensal microbes or incidentally carried pathogens contribute to the resistome.”

6. Most patients received antibiotics prior to sample collection, which presumably would affect ARG expression data. If available, the potential interaction of antibiotic exposure and ARG expression should be examined.

We appreciate this suggestion. Given that 90% of patients from both the adult and the pediatric cohort received antibiotics prior to sample collection (Supplementary Table 1), and given that we only have yes/no data regarding antibiotic administration for these patients, we are unable to meaningfully evaluate this interaction. However, we agree with this reviewer’s comments, and have ongoing work in a new longitudinal cohort to evaluate this specific suggestion in more depth. We also now state this as a limitation in the discussion as follows (line 391): “Third, while an equal proportion (90%) of the pediatric and adult cohorts received antibiotics prior to intubation, the specific antibiotic exposures in terms of exact drugs may have differed, affecting ARG expression. Longitudinal samples to evaluate changes in ARG expression over time, particularly in relation to antibiotic exposure, were not available.”

Minor comments:

7. Fig 4; not sure how “bacterial abundance” has p-value of <0.01 when the aOR=1.

The confidence interval was incredibly small and the aOR was just over 1, but with significant figures this could not be appreciated. Nonetheless, this comment made us revisit each aspect of the model input. We realized that given the similarly high range of bacterial abundance (most samples in the thousands), a logarithmic transformation of the abundance data in the model would be more appropriate. We have thus adjusted the analysis in Figure 4A to use logarithmic bacterial abundance, and the aOR is now 1.17 (95% CI: 1.09-1.25) with a p-value of <0.01 .

8. The cross-sectionality of the cohort and its limitations needs to be addressed.

We thank the reviewer for this comment and appreciate with this point. We have added a line to the “limitations” section of the discussion regarding the lack of availability of longitudinal samples, with the following sentence on line 393: “Longitudinal samples to evaluate changes in ARG expression over time, particularly in relation to antibiotic exposure, were not available.”

REVIEWERS' COMMENTS

Reviewer #1 (Remarks to the Author):

The authors have addressed my comments.

Reviewer #2 (Remarks to the Author):

The authors have largely been responsive to our prior critiques. There were two lingering concerns.

--While we appreciated the discussion of the limitation of unknown antibiotic exposure (line 391-393), we remain concerned that the authors have not fully explained how this limitation affects their interpretation of metatranscriptomic differences. In other words, from their data, it is unclear if the increased expression of ARGs results from expansion of the antibiotic resistance reservoir, enhanced transcription of ARGs, or a relative increase in detection after elimination of sensitive strains. While these questions cannot be experimentally addressed with the current dataset, a clearer discussion of these points would be valuable.

--In the title and introduction, the term 'antibiotic reservoir' is used. It would be very helpful if they could define what their use of this term means relative to transcriptomic data.

REVIEWERS' COMMENTS

Reviewer #1 (Remarks to the Author):

The authors have addressed my comments.

Reviewer #2 (Remarks to the Author):

The authors have largely been responsive to our prior critiques. There were two lingering concerns.

--While we appreciated the discussion of the limitation of unknown antibiotic exposure (line 391-393), we remain concerned that the authors have not fully explained how this limitation affects their interpretation of metatranscriptomic differences. In other words, from their data, it is unclear if the increased expression of ARGs results from expansion of the antibiotic resistance reservoir, enhanced transcription of ARGs, or a relative increase in detection after elimination of sensitive strains. While these questions cannot be experimentally addressed with the current dataset, a clearer discussion of these points would be valuable.

Thank you for this input. To address this point, we have added the following to the limitations section of the discussion:

Line 252: "Third, while an equal proportion (90%) of the pediatric and adult cohorts received antibiotics prior to intubation, the specific antibiotic exposures in terms of exact drugs may have differed, affecting ARG expression. It is possible that specific antibiotic exposures may have influenced our metatranscriptomic results. Longitudinal samples to evaluate changes in ARG expression over time, particularly in relation to antibiotic exposure, were not available and their collection should be prioritized in future studies."

--In the title and introduction, the term 'antibiotic reservoir' is used. It would be very helpful if they could define what their use of this term means relative to transcriptomic data.

We thank the reviewer for this input as well. We have added the following sentence to the introduction to clarify the use of this term in the context of transcriptomic data:

Line 63: "Additionally, while DNA metagenomic sequencing captures the potential ARG reservoir of the lung microbiome, RNA sequencing allows for assessment of the actively transcribed component of this reservoir."